# Prediction of Neoadjuvant Chemotherapy Response in Osteosarcoma Using Convolutional Neural Network of Tumor Center ^18^F-FDG PET Images

**DOI:** 10.3390/diagnostics11111976

**Published:** 2021-10-25

**Authors:** Jingyu Kim, Su Young Jeong, Byung-Chul Kim, Byung-Hyun Byun, Ilhan Lim, Chang-Bae Kong, Won Seok Song, Sang Moo Lim, Sang-Keun Woo

**Affiliations:** 1Radiological & Medico-Oncological Sciences, University of Science & Technology, Seoul 34113, Korea; jingyu8754@kirams.re.kr; 2College of Medicine, University of Ulsan, Seoul 05505, Korea; x-ppul@hanmial.net; 3Department of Nuclear Medicine, Korea Institute of Radiology and Medical Sciences, Seoul 01812, Korea; xikian@kirams.re.kr (B.-C.K.); nmbbh@kirams.re.kr (B.-H.B.); ilhan@kirams.re.kr (I.L.); smlim328@kirams.re.kr (S.M.L.); 4Department of Orthopedic Surgery, Korea Institute of Radiology and Medical Sciences, Seoul 01812, Korea; cbkongmd@gmail.com (C.-B.K.); wssongmd@gmail.com (W.S.S.)

**Keywords:** ^18^F-FDG heterogeneity, convolutional neural network, chemotherapy response, osteosarcoma, machine learning

## Abstract

We compared the accuracy of prediction of the response to neoadjuvant chemotherapy (NAC) in osteosarcoma patients between machine learning approaches of whole tumor utilizing fluorine−^18^fluorodeoxyglucose (^18^F-FDG) uptake heterogeneity features and a convolutional neural network of the intratumor image region. In 105 patients with osteosarcoma, ^18^F-FDG positron emission tomography/computed tomography (PET/CT) images were acquired before (baseline PET0) and after NAC (PET1). Patients were divided into responders and non-responders about neoadjuvant chemotherapy. Quantitative ^18^F-FDG heterogeneity features were calculated using LIFEX version 4.0. Receiver operating characteristic (ROC) curve analysis of ^18^F-FDG uptake heterogeneity features was used to predict the response to NAC. Machine learning algorithms and 2-dimensional convolutional neural network (2D CNN) deep learning networks were estimated for predicting NAC response with the baseline PET0 images of the 105 patients. ML was performed using the entire tumor image. The accuracy of the 2D CNN prediction model was evaluated using total tumor slices, the center 20 slices, the center 10 slices, and center slice. A total number of 80 patients was used for k-fold validation by five groups with 16 patients. The CNN network test accuracy estimation was performed using 25 patients. The areas under the ROC curves (AUCs) for baseline PET maximum standardized uptake value (SUVmax), total lesion glycolysis (TLG), metabolic tumor volume (MTV), and gray level size zone matrix (GLSZM) were 0.532, 0.507, 0.510, and 0.626, respectively. The texture features test accuracy of machine learning by random forest and support vector machine were 0.55 and 0. 54, respectively. The k-fold validation accuracy and validation accuracy were 0.968 ± 0.01 and 0.610 ± 0.04, respectively. The test accuracy of total tumor slices, the center 20 slices, center 10 slices, and center slices were 0.625, 0.616, 0.628, and 0.760, respectively. The prediction model for NAC response with baseline PET0 texture features machine learning estimated a poor outcome, but the 2D CNN network using ^18^F-FDG baseline PET0 images could predict the treatment response before prior chemotherapy in osteosarcoma. Additionally, using the 2D CNN prediction model using a tumor center slice of ^18^F-FDG PET images before NAC can help decide whether to perform NAC to treat osteosarcoma patients.

## 1. Introduction

Osteosarcoma is the most common primary malignant bone tumor, typically occurring in the metaphysis of the long bones and occurs mainly between the ages of 15 and 25, and occurs more frequently in men than in women [1]. For most of the 20th century, the 5-year survival rate of osteosarcoma was as low as 20% [2]. Application of neoadjuvant chemotherapy (NAC) therapy significantly improves long-term survival in patients with high-grade osteosarcoma. Recently, the NAC protocol has been included before and after surgery for osteosarcoma patients [3]. However, NAC for osteosarcoma has a toxicity and ineffective problem [4,5,6]. Ineffective chemotherapy can cause drug resistance [7] and delayed tumor removal surgery can compromise clinical outcomes [8]. Therefore, predicting the histological response to NAC and determining whether to maintain treatment is important for managing osteosarcoma patients.

Tumor necrosis rate is a criterion for evaluating the chemotherapy response evaluation [9] and has been evaluated as the most important prognostic factor in osteosarcoma [10], but it has a limitation that was hard to predict before NAC and can be evaluated only in the resected specimen after completing NAC. To overcome this limitation, the evaluation of the chemotherapy response for osteosarcoma using computed tomography (CT) [11], magnetic resonance imaging (MRI) [7,12,13], and ^18^F-fluoro-2-deoxy-D-glucose positron emission tomography (^18^F-FDG PET) [14,15,16] has been studied. For prediction of the histological response to NAC before surgery, assessing the tumor volume changes in sequential MRI was used [7,12]. However, in these studies, regression and cystic degeneration of the tumor osteoid matrix by prior chemotherapy occurred slowly in the responding group. The change in tumor volume and histological results in MRI before and after prior chemotherapy was inconsistent. Nuclear medicine imaging using ^18^F-FDG PET is mainly used to determine the diagnosis and staging of cancer patients [17]. Standard uptake value (SUV) is a quantification factor that can be applied in various ways in various cancers. In addition, metabolic tumor volume (MTV) and total lesion glycolysis (TLG) are used to diagnose cancer patients and predict prognosis [18,19]. ^18^F-FDG PET is a functional imaging method based on increased glucose usage of malignant cells, so it can detect changes in tissue metabolism that precede structural changes, so it has been reported to be useful for predicting clinical outcomes or evaluating chemotherapy responses in osteosarcoma [14,15]. Recent studies with osteosarcoma patients reported that metabolic tumor volume (MTV) and total lesion glycolysis (TLG) obtained from ^18^F-FDG PET after one cycle of chemotherapy can predict the response of chemotherapy [16,20]. However, in these studies, metabolic tumor volume (MTV) and total lesion glycolysis (TLG) obtained from ^18^F-FDG PET prior to chemotherapy could not predict the response of chemotherapy.

Image texture features from ^18^F-FDG PET contain information about the cell conditions or behaviors. Each image texture feature represents the cell volume, cell size, cell surface texture, glucose uptake, and so on. The prediction models with these image texture features can predict more accuracy than the prediction model with images without any pre-analysis [21].

The deep learning techniques have been used to estimate the prediction model with a DNA sequence promoter binding site and amino acid embedding representation [22,23]. Research results of applying a 2-dimensional convolutional neural network (2D CNN), one of these deep learning techniques, to MRI images of brain tumor patients have been published [24,25]. Additionally, a study that predicted the response of prior chemotherapy in esophageal cancer by applying the deep learning to ^18^F-FDG PET images has also been published [26].

In previous studies, it was confirmed that the use of intertumoral heterogeneity factors (such as MTV and TLG) extracted from ^18^F-FDG PET images obtained after one cycle of NAC improves the prognostic performance of NAC in osteosarcoma patients [16,20]. However, these studies did not analyze MTV and TLG, which are heterogeneous factors in tumors extracted from ^18^F-FDG PET images obtained before NAC. According to previous reports, ^18^F-FDG tumor heterogeneity holds promise for predicting chemotherapy response and 2D CNN is a state-of-the-art method for this prediction.

In this study, the NAC prediction model was estimated using image texture features of ^18^F-FDG PET images from osteosarcoma patients before and after NAC with the machine learning and deep learning algorithm. The performance of predictive models according to the intratumor region was estimated with various intratumor regions as input in a 2D CNN network.

## 2. Materials and Methods

### 2.1. ^18^F-FDG PET/CT

The retrospective study was conducted in a cohort of 81 osteosarcoma patients who were diagnosed at the Korea Institute of Radiology and Medical Sciences from June 2006 to May 2014. Each ^18^F-FDG PET image was obtained before and after the first NAC. The duration of ^18^F-FDG PET before treatment (baseline PET0) and the onset of the first NAC was less than two weeks. An ^18^F-FDG PET image was taken within two to three weeks at the end of the first NAC (after NAC) [15].

All osteosarcoma patients received NAC (during four weeks) involving a combination of methotrexate (a dose of 8–12 g/m^2^), adriamycin (a dose of 60 mg/m^2^), and cisplatin (a dose of 100 mg/m^2^) at intervals of three weeks. The surgery was performed three weeks after the end of the second NAC [15]. The NAC response was evaluated based on the tumor by a pathologist. Tumor necrosis percentages of Grades III and IV (necrosis of 90% or more) indicated a good response, and Grades I and II (less than 90% necrosis) indicated a poor response [9]. A total of 105 osteosarcoma patients were classified as responders (*n* = 47) and non-responders (*n* = 58). The detailed research subject information is presented in Table 1.

For each patient, a ^18^F-FDG PET/CT scan was acquired before NAC and after NAC using a Biograph 6 PET/CT scanner (Siemens Medical Solutions, Erlangen, Germany). PET scan was performed at 3.5 min/frame in the 3-dimensional (3D) model, 60 min after 7.4 MBq/kg ^18^F-FDG was injected intravenously. PET/CT images were reconstructed using CT for attenuation correction (field-of-view, 680 m × 680 m; voxel size, 4 m × 4 m × 3 m) and 3D ordered subset expectation maximization algorithms. The information on image texture features is presented in Table 2.

### 2.2. Quantitative Analysis of ^18^F-FDG Uptake Heterogeneity

The ^18^F-FDG uptake heterogeneity features were calculated using the Local Image Features Extraction (LIFEx) version 4.0 software package [27]. To include all tumor regions in the ^18^F-FDG PET, we defined the region growing method based on SUV ≥1.5 [28].

We computed the quantitative texture features (i.e., gray-level co-occurrence matrix, gray level run-length matrix, gray-level neighborhood intensity-difference matrix, and gray level size-zone matrix) to investigate the ^18^F-FDG heterogeneity within the tumor. Additionally, we calculated the conventional ^18^F-FDG features (i.e., the SUVmax, MTV, and TLG). Quantitative texture features and conventional ^18^F-FDG features were calculated using LIFEx.

Random forest and support vector machine (SVM) algorithms were used to classify the treatment response of osteosarcoma patients. To achieve this goal, the ratio of machine learning training data to test data was set as 7:3. Cross-validation was performed 10 times to increase the statistical reliability of the performance measurements.

### 2.3. Convolutional Neural Network

A 2D CNN assumes that the inputs have a geometric relationship such as rows and columns in images [23]. PyTorch 1.9.0+cu102 was used for deep learning and the whole scripts were written in Python 3.8.6. The input layer of the 2D CNN produces a convolution of a small image, known as a feature map. The feature map is generated by a filter that is moved across the input image. From this feature map, values are extracted and used as input for the pooling layer. In this study, we designed the 2D CNN as shown in Figure 1.

The 2D CNN worked in 2D convolutional layers with numerous slices of tumor volume in the ^18^F-FDG PET images. The convolutional layer filter size was 5 × 5, and the numbers of filters were 32 in both the first and second convolutional layers as well as in the max-pooling method, using a 2 × 2 filter in the pooling layer. In the activation function, we used the rectifier linear unit (ReLu); we calculated the loss based on softmax, cross-entropy and used adaptive moment estimation (Adam) for loss optimization. To avoid overfitting with the training dataset, we implemented the dropout technique after both the first and second fully connected layers [29].

To evaluate the accuracy of the 2D CNN prediction model, slides from the tumor were used. Eighty patients for k-fold validation were separated into five groups, each group containing 16 patients, and consisting of the training and validation set. Four groups were used for training and one group was used for the validation test dataset. The k-fold cross-validation was performed five times with the group of separated patients. A total of 640 slices from 64 patients (10 slices from tumor center, 64 patients from four groups) were used for the training set and 160 slices from 16 patients (10 slices from tumor center) were used for the validation set. Deep learning test processing consisted of 640 slices of the training dataset from 10 slices of 64 patients, and we added 25 slices of the test dataset from center 10 slices and center slice.

### 2.4. Statistical Analysis

Significant quantitative features of ^18^F-FDG homogeneity for the prediction of the NAC response were assessed using receiver operating characteristic (ROC) curve analysis with 95% confidence intervals (95% CIs). Statistical significance was confirmed using logistic regression analysis, with *p*-values < 0.05. To compare the AUCs between the 2D CNN and ^18^F-FDG heterogeneity, we performed independent t-tests. All statistical analysis was performed in MedCalc version 18.6 (MedCalc Software bvba, Mariakerke, Belgium).

## 3. Results

### 3.1. ^18^F-FDG Quantitative Analysis

^18^F-FDG PET images of the responder and non-responder are shown in Figure 2. Based on quantitative feature analysis, PET1 features had a higher ROC-AUC value loss optimizer than the baseline PET0 (Table 3). The highest AUC for ^18^F-FDG uptake heterogeneity in baseline PET0 was obtained using the gray level size zone matrix (GLSZM), a feature reflecting the intensity size zone matrix in ^18^F-FDG PET images. The highest AUC in PET1 was obtained for the standardization of SUV (SUV_SD).

### 3.2. Quantitative ^18^F-FDG Heterogeneity Features

Forty-seven features in the T-SNE plot of 105 patients in Figure 3 are shown for the identification of the distribution of non-responder/responder osteosarcoma patients. The accuracy of the prediction model with random forest and support vector machine was calculated using the total image texture features. The ROC-AUC values of baseline PET0 maximum standardized uptake value (SUVmax), total lesion glycolysis (TLG), and metabolic tumor volume (Volume) were 0.532 (*p*-value: 0.622), 0.507 (*p*-value: 0.918), and 0.510 (*p*-value: 0.881), respectively (Table 4). Analysis of baseline PET0 ^18^F-FDG uptake heterogeneity features yielded a ROC-AUC for GLSZM of 0.626 (*p*-value: 0.045) (Figure 4).

The ROC-AUC values of PET1 SUVmax, TLG, and Volume were 0.793, 0.764, and 0.741, respectively (Table 4). These values were significantly different between responders and non-responders (all *p*-values < 0.001). Analysis of PET1 ^18^F-FDG uptake heterogeneity features demonstrated a ROC-AUC for GLSZM of 0.741 (*p*-value: < 0.001) (Figure 5).

The sensitivity, specificity, AUC, train accuracy, and test accuracy of the prediction for chemotherapy response in Table 3 were calculated using the random forest algorithm and the SVM algorithm. The random forest algorithm prediction and support vector machine for test accuracy using a total of 47 text features were 0.55 and 0.54, respectively.

### 3.3. Predictive Accuracies of ^18^F-FDG PET 2D CNN

As shown in Figure 6, after dimension reduction, the fully connected layers were separated into two classes. In the two cases, the classes were clearly separated. We obtained a relatively high precision rate for the chemotherapy response.

The training set accuracy of fold1, fold2, fold3, fold4, and fold5 in k-fold validation was 0.968 ± 0.01. The test validation set accuracy was 0.610 ± 0.03. The loss function and train/test accuracy graph in k-fold validation were estimated by each step. The results of the test set accuracy for the neoadjuvant chemotherapy response prediction deep learning model are presented in Table 5. The training accuracy of total tumor slices, the center 20 slices, center 10 slices, and center slices were 0.984, 0.983, 0.966, and 0.988, respectively. The validation accuracy of training accuracy of total tumor slices, the center 20 slices, center 10 slices, and center slices were 0.625, 0.616, 0.628, and 0.760, respectively. The loss function and train/test accuracy graph in the test set were estimated.

## 4. Discussion

In this study, we investigated and validated the accuracy of using a 2D CNN trained on ^18^F-FDG data or using FDG uptake heterogeneity features for predicting response to NAC. Before NAC, only GLSZM (AUC = 0.626, sensitivity = 0.579, specificity = 0.721, *p*-value = 0.045), an ^18^F-FDG uptake heterogeneity feature reflecting the image intensity size zone, could predict the NAC response, while SUVmax (AUC = 0.532, sensitivity = 0.842, specificity = 0.302, *p*-value = 0.622), TLG (AUC = 0.507, sensitivity = 0.763, specificity = 0.395, *p*-value = 0.918), and MTV (AUC = 0.510, sensitivity = 0.816, specificity = 0.349, *p*-value = 0.881) could not; this prediction result is similar to the results of previous studies [16,20]. ^18^F-FDG PET heterogeneity features of data collected after NAC could predict the chemotherapy response (see Table 3 and Table 4). Likewise, the 2D CNN had good predictive accuracy before NAC (AUC = 0.920, sensitivity = 0.965, specificity = 0.881), which increased after NAC (AUC = 0.955, sensitivity = 0.983, specificity = 0.927). There were no statistically significant differences in the predictive accuracies of the ^18^F-FDG PET 2D CNN before and after NAC (*p*-value = 0.158). Since the accuracy of using a 2D CNN trained on ^18^F-FDG data for predicting a response to NAC was much better than the accuracy of using FDG uptake heterogeneity features, we verified these results using validation data from 25 patients.

Recently, machine learning and deep learning techniques have been applied to pattern recognition in medical images [30]. With the development of computer hardware and the growth in medical imaging data, the application of deep learning technology for computer-aided diagnosis (CAD) in medical imaging has recently been a popular research topic. This technique uses deep artificial neural networks to learn the image shape patterns of the objects of interest based on a large training dataset. Deep learning has a better performance than existing machine learning methods in object detection and classification. In addition, the use of deep learning is increasingly being used for medical image analysis [31].

Machine learning and deep learning techniques have been applied in various studies by developing technologies of machine learning and deep learning. Deep learning approaches have most commonly been applied in MR studies [32]. This preliminary study had several important findings. A total of 47 image features were extracted from the ^18^F-FDG PET/CT images. Imaging features related to the chemotherapy response were identified using the AUC value. The AUC values of all the image texture features were similar to about 0.5. The test accuracy of the prediction model using the total image texture features and random forest and support vector machine was similar at 0.55 and 0.54, respectively. A t-SNE plot analysis was performed to identify the distribution of image texture features and images from patients. As a result, it was determined that the prediction model using the AUC of image texture features, machine learning model, and t-SNE plot could not distinguish between the responders and non-responders.

^18^F-FDG heterogeneity features, gray-level co-occurrence matrix, gray-level run-length matrix, gray level neighborhood intensity-difference matrix, gray level size zone matrix as well as intensity features were calculated using Lifex software [20,32]. This quantitative analysis method was used in a previous study to predict the NAC response in breast cancer patients [33,34], and survival in oropharyngeal cancer [35] and pancreatic ductal adenocarcinoma patients [36].

Previous studies have reported that a 2D CNN based on ^18^F-FDG had a higher accuracy for predicting response, but did not compare this predicting response with the accuracy of using FDG heterogeneity features [26,37], which made it difficult to understand the source of the increased accuracy obtained using the 2D CNN. Cheng et al. showed that the diagnosis prediction model with ^18^F-FDG PET/CT image texture features from lung cancer was 0.87–0.92 with AUC as a classical method and 0.91 with the CNN model [35] and Ypsilantis et al. showed the accuracy of predicting response to neoadjuvant chemotherapy with PET image texture features from esophageal cancer was 73.4 ± 5.3 with 3S-CNN and 66.4 ± 5.9 with 1S-CNN [24,26].

Another previous study visually represented the convolutional layers of the feature map in a 2D CNN. This 2D CNN revealed that the first convolutional layer extracted edge and blob features, which are relatively simple image features. The second convolutional layer extracted the related texture features [38,39,40].

Based on the convolutional layer characteristics, we assessed the correlation between the accuracy of using a 2D CNN and that of using ^18^F-FDG heterogeneity features. We found that the NAC prediction accuracy of the 2D CNN model depended on the AUCs of the intensity and heterogeneity features; the change in accuracy for baseline PET0 and PET1 was 1.47- and 1.29-fold, respectively. According to the ROC curve analysis, the sensitivity of the 2D CNN model, before and after NAC, did not significantly change (0.965 to 0.983). However, the specificity significantly changed from 0.881 to 0.927. This is because it is possible to predict the non-response to response more accurately after observing the effect of NAC. The prediction model using 2D CNN showed a more accurate result in the prediction model to predict responders and non-responders, although the prediction model using machine learning and AUC showed poor prediction results.

The predictive accuracy of the 2D CNN was affected by its deep learning architecture. Before training the 2D CNN, we optimized the 2D CNN architecture using the grid-search technique [39]. Based on the optimized 2D CNN architecture, we confirmed two convolutional layers with a 5 × 5 filter. Consequently, the 2D CNN architecture included two convolutional and two fully connected layers, which were similar to a previously reported ^18^F-FDG PET 2D CNN architecture [26]. In this study, we performed the k-fold cross-validation and included a dropout layer in the 2D CNN model to avoid overfitting the training data; this approach is widely used in applied deep learning techniques [41].

It was identified that the accuracy was higher using 10 center slices than a single-center slice by comparing the accuracy of the 2D CNN prediction model using 10 center slices and a single-center slice obtained from tumors. The accuracy of 10 slices and single slice were 0.628 and 0.760, respectively. In this study, the 2D CNN predictive model using a single slice was higher than that of 10 slices, but was not completely reliable due to the small size of the patient group in the experiment. In the future, it is necessary to study the relationship between the number of tumor slices and the accuracy of the predictive model by analyzing tumors obtained from more patients.

It is difficult to apply this to clinical practice because many patients are required for an accurate deep learning prediction model, although the test accuracy of the deep learning prediction model is high. Applying gene expression factors to machine learning predictive models can yield higher test accuracy. Radiogenomics is a field of study that explores and uses the relationship between nuclear image analysis and gene expression. In many studies, the relationship between gene expression and image texture features has been found using radiogenomics techniques, and predictive models were estimated. If the radiogenomics technique is applied to the predictive model to discriminate chemotherapy responders, improved test accuracy could be obtained.

This study had some limitations. First, only patients who met the criteria were selected from the cohort of consecutively treated patients and retrospectively analyzed. Second, data from a small group of patients collected from one institution were analyzed for this study. To achieve reliability of the results, multi-center cross-validation should be performed using large patient datasets from various institutions.

## 5. Conclusions

The prediction model using the machine learning algorithm has been used to estimate poor outcome for NAC in osteosarcoma, but the 2D CNN prediction model using ^18^F-FDG PET images before NAC can predict the treatment response prior to chemotherapy in osteosarcoma. Additionally, the performance of a prediction model evaluation was different depending on the intratumor region applied to the 2D CNN network. The 2D CNN prediction model using tumor center ^18^F-FDG PET images before NAC can be helpful in deciding whether to perform NAC in the treatment of osteosarcoma patients.

## Figures and Tables

**Figure 1 diagnostics-11-01976-f001:**
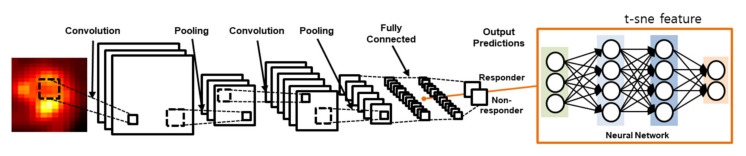
The ^18^F-FDG 2D CNN model for predicting the response to neoadjuvant chemotherapy. The 2D CNN model consisted of two convolution layers and two fully connected layers.

**Figure 2 diagnostics-11-01976-f002:**
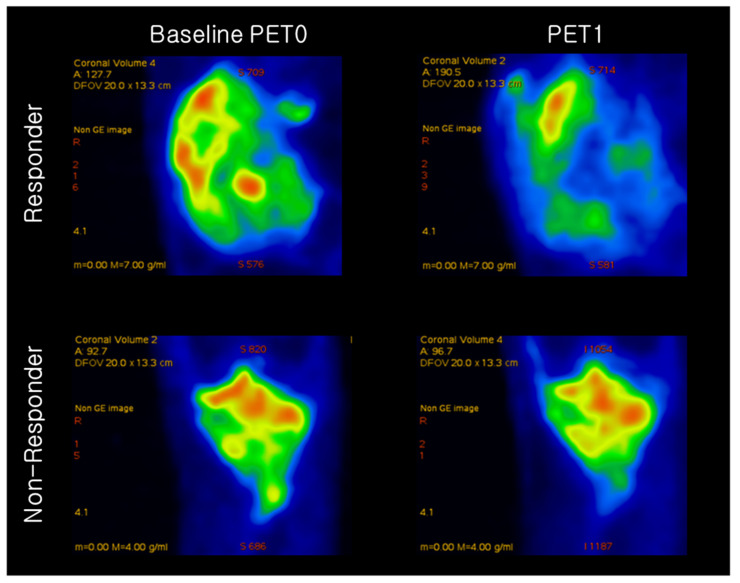
Representative ^18^F-FDG PET image of osteosarcoma in a responder and non-responder to neoadjuvant chemotherapy. Responder had SUVmax values of 11.33 and 4.43 at baseline PET0 and after neoadjuvant chemotherapy (PET1), respectively. Non-responder had SUVmax values of 5.62 and 3.21 at baseline PET0 and after neoadjuvant chemotherapy (PET1), respectively.

**Figure 3 diagnostics-11-01976-f003:**
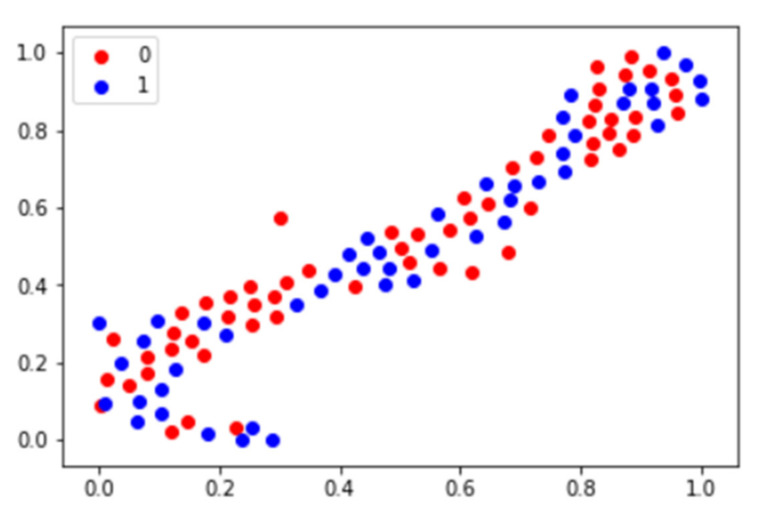
T-SNE plot using image texture features of osteosarcoma patients. In the plot, 0 represents the chemotherapy non-responder and 1 represents the chemotherapy responder.

**Figure 4 diagnostics-11-01976-f004:**
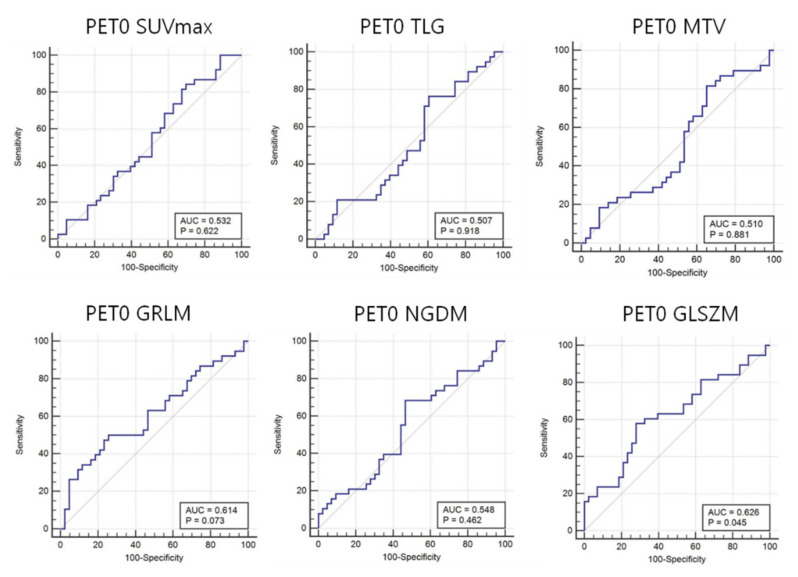
Area under the receiver operating characteristic curves (AUC) for ^18^F-FDG heterogeneity features in baseline PET0. Conventional parameters (i.e., maximum standardized uptake value (SUVmax), total lesion glycolysis (TLG), and metabolic tumor volume (MTV)), cannot predict the response to neoadjuvant chemotherapy before treatment. In contrast, the ^18^F-FDG intensity size zone feature (gray-level size zone matrix: GLSZM) heterogeneity can predict this response.

**Figure 5 diagnostics-11-01976-f005:**
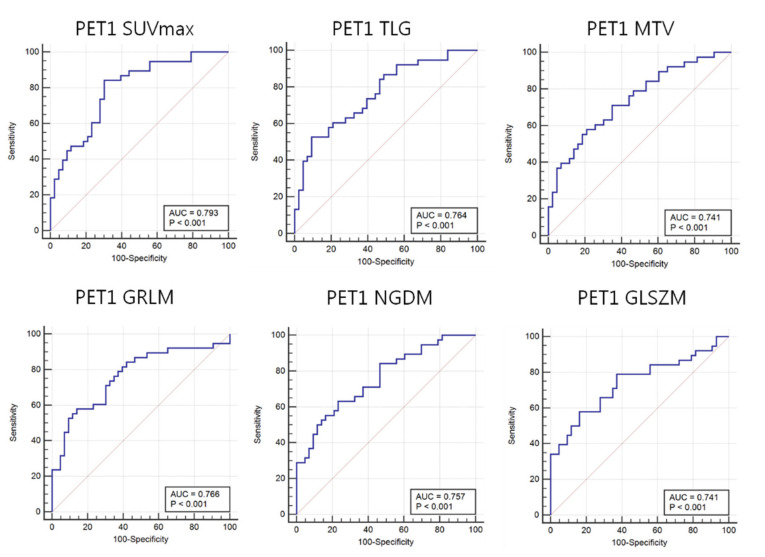
Area under the receiver operating characteristic curves (AUC) for ^18^F-FDG heterogeneity features in PET1. Maximum standardized uptake value (SUVmax), total lesion glycolysis (TLG), and metabolic tumor volume (MTV) as well as ^18^F-FDG uptake heterogeneity features such as image voxel alignment heterogeneity (GLRIM_HGHGE), image neighborhood intensity difference (NGLDM_SNE), and image intensity size zone (GLSZM) can predict the response to neoadjuvant chemotherapy.

**Figure 6 diagnostics-11-01976-f006:**
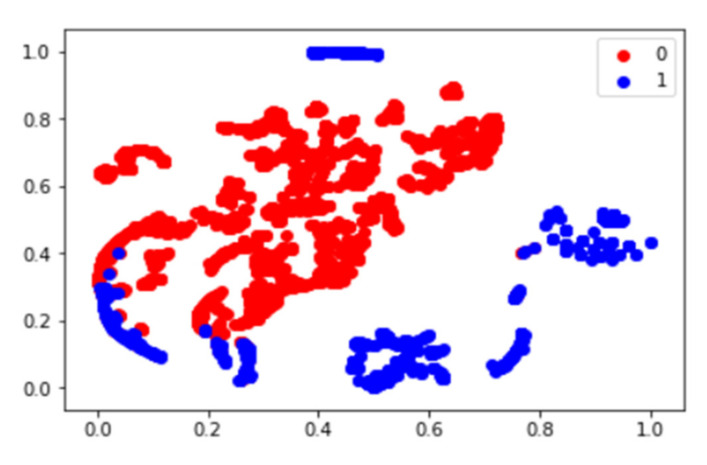
Deep features T-SNE plot using patients of osteosarcoma baseline PET0. In the plot, 0 represents the chemotherapy non-responder and 1 represents the chemotherapy responder.

**Table 1 diagnostics-11-01976-t001:** Information on training and validation subjects with osteosarcoma who responded to neoadjuvant chemotherapy.

Characteristics	Value
Sex, *n* (%)	
Female	30 (29.50%)
Male	75 (70.50%)
Age, *n* (%)	
years ≤ 19	80 (77.14%)
years >19	25 (22.86%)
Location of primary tumor, *n* (%)	
Femur	59 (56.19%)
Tibia	35 (33.33%)
Fibula	5 (4.76%)
Humerus	4 (3.80%)
Pelvis	2 (1.92%)
AJCC stage, *n* (%)	
IIA	37 (35.23%)
IIB	64 (60.95%)
III	2 (1.91%)
IV	2 (1.91%)
Pathologic subtype, *n* (%)	
OB (Osteoblastic)	78 (74.28%)
CB (Chondroblastic)	13 (12.38%)
FB (Fibroblastic)	7 (6.67%)
Others	7 (6.67%)
Histologic response, *n* (%)	
Responder	47 (45.76%)
Non-responder	58 (54.24%)

**Table 2 diagnostics-11-01976-t002:** Index of textural features in global, local, and regional areas.

Feature Family	Features
Intensity histogram	SUVmax
	SUVmean
	Standard deviation (SUV_SD)
	Total lesion glycolysis (TLG)
	Metabolic tumor volume (MTV)
	1^st^ entropy
Gray level co-occurrence matrix (GLCM)	Energy
	Contrast
	Entropy
	Homogeneity
	Dissimilarity
Neighboring gray level dependence matrix(NGLDM)	Contrast
	Coarseness
	Busyness
	SNE (Small number emphasis)
Gray level run length matrix(GLRLM)	SRE (Short run emphasis)
	LRE (Long run emphasis)
	GLNU (Gray level non-uniformity)
	RLNU (Run length non-uniformity)
	SRLGE (Low gray level run emphasis)
	SGHGE (High gray level run emphasis)
Gray level size zone matrix(GLSZM)	SAE (Small zone emphasis)
	LAE (Large zone emphasis)
	GLN (Gray level non-uniformity)
	SZN (Zone size non-uniformity)
	LGLZE (Low gray level zone emphasis)
	HGLZE (High gray level zone emphasis)

**Table 3 diagnostics-11-01976-t003:** Random forest and support vector machine accuracy performed on total image texture features from 105 osteosarcoma patients in baseline PET0.

Chemotherapy Response	Random Forest	Support Vector Machine
Sensitivity	0.53	0.75
Specificity	0.61	0.83
Precision	0.54	0.57
Dice coefficient	0.49	0.48
AUC	0.55	0.52
Accuracy	0.55	0.54

**Table 4 diagnostics-11-01976-t004:** The area under the receiver operating characteristic curve for ^18^F-FDG uptake heterogeneity features.

Features	Discrimination	Baseline PET0		PET1	
		AUC	*p-*Value	AUC	*p-*Value
SUV_max	Intensity	0.532	0.622	0.793	<0.001
SUV_SD	Intensity	0.505	0.940	0.802	<0.001
TLG	Intensity	0.507	0.918	0.764	<0.001
Volume	Shape	0.510	0.881	0.741	<0.001
GLRLM_SGHGE	Voxel-alignment	0.614	0.073	0.766	<0.001
NGLDM_SNE	Neighborhood intensity difference	0.548	0.462	0.757	<0.001
GLSZM_HGLZE	Intensity size zone	0.626	0.045	0.741	<0.001
GLCM_entropy	Normalized Co-occurrence matrix	0.588	0.165	0.744	<0.001

SUVmax, maximum standardized uptake value; TLG, total lesion glycolysis; MTV, metabolic tumor volume; GLRLM_SGHGE, Gray level run length matrix_High gray level run emphasis; NGLDM_SNE, Neighboring gray level dependence matrix_Small number emphasis; GLSZM_HGLZE, Gray level size zone matrix_High gray level zone emphasis; GLCM_entropy, Gray-level co-occurrence matrix_Entropy; AUC, area under the receiver operating characteristic curve.

**Table 5 diagnostics-11-01976-t005:** The accuracy of test set for neoadjuvant chemotherapy response prediction deep learning model.

2D CNN	Total Tumor Slices	Center 20 Slices	Center 10 Slices	Center Slice
Train accuracy	0.984	0.983	0.966	0.988
Test accuracy	0.625	0.616	0.628	0.76

## Data Availability

Data sharing not applicable.

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
