# Peer review of "Prediction of Neoadjuvant Chemotherapy Response in Osteosarcoma Using Convolutional Neural Network of Tumor Center 18F-FDG PET Images"

_diagnostics, 2021, doi:10.3390/diagnostics11111976_

Round 1
Reviewer 1 Report
In this study, Kim et al. proposed convolutional neural network (CNN) architecture to predict neoadjuvant chemotherapy response in osteosarcoma. The model input is fluorine-18 fluorodeoxyglucose (18F-FDG) and the authors extracted different features for prediction purposes. However, there are some major points in this study:
1. English language should be improved.
2. In some cases, the performance results look too low with AUC at about 0.5.
3. The authors should spend one paragraph in the introduction to explain the gap of research.
4. p-values should be shown in Table 1.
5. In Table 1, I didnot see the authors separated data into training and validation.
6. The authors should explain more about the feature names.
7. How did the authors deal with hyperparameter tuning? When comparing the performance among different models, are they tuned to optimal ones? It should be clear to have a fair comparison.
8. First paragraph in "Results" should be excluded.
9. Deep learning, especially CNN, has been used in previous studies i.e., PMID: 31920706 and PMID: 31750297. Therefore, the authors are suggested to refer to more works to attract a broader readership.
10. t-SNE in Fig. 3 shows nothing since it cannot be used to separate two samples.
11. Low performance (AUC) in Fig. 4.
12. Source codes should be provided for replicating the methods.
13. The authors should compare the predictive performance to previous studies on the same problem/data.
Reviewer 2 Report
The paper entitled „Predication of neoadjuvant chemotherapy response in osteosar coma using convolutional neural network of tumor center 18F- 3 FDG PET images” by Kim et. al deals with a classic problem by comparison of accuracy metric in case of neoadjuvant chemotherapy 17 (NAC) in osteosarcoma patients between fluorine-18 fluorodeoxyglucose and a convolutional neural network of intratumor image region.
In this study is based on 105 patients with osteosarcom, features extracted from (PET/CT) images were calculated using LIFEX version 4.0.
The paper is well structured, the introduction section is based on 23 references, all are in according with the treated subject.
In tables 1and 2 are shown in detail the subjects with osteosarcoma and textural features, respectively.
For the evaluating accuracy besides Random forest and support vector machine (SVM) algorithms the convolutional neural network were used.
A few minor observations:
- Please to add in the introduction part the paper structure.
- In section 2.3. Convolutional neural network, please add the used software in implementing of the network.
- Please to improve the quality of the figures 3 and 4.
- Please to specify the time consuming for Convolutional neural network.
- #247 line: the authors said “The training set accuracy of fold1, fold2, fold3, fold4 and fold5 in k-fold validation was 0.968±0.01” please to specify what mean the second value ±0.01
- Besides Sensitivity, Specificity, Accuracy computed with Random forest and support vector machine , please to add the significant metrics as precision and Dice coefficient.
- In Discussion section, the authors reported their research to the papers 23, 34, please add for these the obtained accuracy.
Author Response
Comments and Suggestions for Authors
The paper entitled „Predication of neoadjuvant chemotherapy response in osteosarcoma using convolutional neural network of tumor center 18F- 3 FDG PET images” by Kim et. al deals with a classic problem by comparison of accuracy metric in case of neoadjuvant chemotherapy 17 (NAC) in osteosarcoma patients between fluorine-18 fluorodeoxyglucose and a convolutional neural network of intratumor image region.
In this study is based on 105 patients with osteosarcoma, features extracted from (PET/CT) images were calculated using LIFEX version 4.0.
The paper is well structured, the introduction section is based on 23 references, all are in according with the treated subject.
In tables 1 and 2 are shown in detail the subjects with osteosarcoma and textural features, respectively.
For the evaluating accuracy besides Random forest and support vector machine (SVM) algorithms the convolutional neural network were used.
A few minor observations:
- Please to add in the introduction part the paper structure.
Respose: Image texture features from 18F-FDG PET have information of cell condition or be-havior. Each image texture features represent cell volume, cell size, cell surface texture, glucose uptake and so on. The prediction model with these image texture features can predict more accuracy than prediction model with images without any pre-analysis.
- In section 2.3. Convolutional neural network, please add the used software in implementing of the network.
Respose: Thank you for your kind mention. The information of software was added in the section 2.3 as follow;
“PyTorch 1.9.0+cu102 was used for deep learning and whole program is written in Python 3.8.6.” in line 156-157, section 2.2
- Please to improve the quality of the figures 3 and 4.
Respose: Thank you for your kind mention. We improved the quality of figures
- Please to specify the time consuming for Convolutional neural network.
Respose: Thank you for your kind mention. The time consuming for Convolution neural network is approximately 11 minutes. (11 minutes and 28 seconds, 11 minutes and 9 seconds, 11 minutes and 17 seconds.) all time was represent by average and standard variation (3 times).
- #247 line: the authors said “The training set accuracy of fold1, fold2, fold3, fold4 and fold5 in k-fold validation was 0.968±0.01” please to specify what mean the second value ±0.01
Respose: Thank you for your kind mention. It represents the average ± standard deviation of accuracy from five fold.
- Besides Sensitivity, Specificity, Accuracy computed with Random forest and support vector machine, please to add the significant metrics as precision and Dice coefficient.
Chemotherapy response |
Random Forest |
Support Vector Machine |
Sensitivity |
0.53 |
0.75 |
Specificity |
0.61 |
0.83 |
Precision |
0.54 |
0.57 |
Dice coefficient |
0.49 |
0.48 |
AUC |
0.55 |
0.52 |
Accuracy |
0.55 |
0.54 |
Respose: Thank you for your kind mention. precision and Dice coefficient was added in the table 4.
- In Discussion section, the authors reported their research to the papers 23, 34, please add for these the obtained accuracy.
Respose: Thank you for your kind mention. We add the obtained accuracy from previous study in discuss section.
“H Wang, et al shows the diagnosis prediction model with 18F-FDG PET/CT image texture features from lung cancer was 0.87~0.92 with AUC as a classical method and 0.91 with CNN model [35] and Petros-Pavlos Ypsilantis et al showed accuracy of predicting re-sponse to neoadjuvant chemotherapy with PET Image texture features from esophageal cancer was 73.4±5.3 with 3S-CNN and 66.4±5.9 with 1S-CNN [24].” In line 328-333
Round 2
Reviewer 1 Report
My previous comments have been addressed well.
Author Response
Response: Thank you for your kind mention. We fixed the errors as follow
All “K-fold” to “k-fold”
Line 2 “Predication” to “prediction”
Line 6 “Sang Moo Lim3” to “Sang Moo Lim3”
Line 5, 15 add the description for ‘co-author’ mark and “These authors contributed equally to this work.”
Line 16 “phone number was removed.
Line 17 “between fluorine-18fluorodeoxyglucose (18F-FDG) uptake heterogeneity features machine learning of whole tumor and a convolutional neural network of intratumor image region” to “between machine learning approaches of whole tumor utilizing fluorine-18fluorodeoxyglucose (18F-FDG) uptake heterogeneity features and a convolutional neural network of the intratumor image region”
Line 31 “metabolic tumor volume (MTV)” to “metabolic tumor volume (MTV),”
Line 38 “2D CNN” to “the 2D CNN”
Line 39 “model using tumor” to “model using a tumor”
Line 52 “have” to “has”
Line 63 “18F-FDG” to “18F-FDG PET”
Line 81 “condition or behavior.” to “conditions or behaviors.”
Line 91 “a deep learning” to “the deep learning”
Line 94 “one cycle NAC” to “one cycle of NAC”
Line 101 “NAC with machine learning” to “NAC with the machine learning”
Line 102 “deep learning” to “the deep learning”
Line 102 “Performance” to “The performance”
Line 103 “were” to “was”
Line 117 “evaluated on the basis of” to “based on”
Line 155 “whole program is” to “the whole scripts were”
Line 174 “from tumor” to “from the tumor”
Line 175 “separated 5 groups” to “separated into 5 groups”
Line 177 “validation” to “the validation”
Line 179 “(10 slice from tumor center) was used for validation set.” to “(10 slices from tumor center) were used for the validation set.
Line 181 “640 slices training data” to “640 slices of training data”
Line 219 “for identification of distribution” to “for identification of the distribution”
Line 220 “The accuracy of prediction model” to “The accuracy of the prediction model”
Line 244 “table 4” to “Table 4”
Line 273 “for neoadjuvant chemotherapy” to “for the neoadjuvant chemotherapy”
Line 274 “was” to “were”
Line 293 “table 3 & table 4” to “Table 3 & Table 4”
Line 306 “in object detection and classification and is increasingly being used for medical image analysis” to “in object detection, classification. In addition, usage of the deep learning is increasingly being used for medical image analysis”
Line 332 “showed accuracy” to “showed the accuracy”
Line 385 “using machine learning to” to “using the machine learning”